# Learning to Learn by Zeroth-Order Oracle

**Yangjun Ruan[1], Yuanhao Xiong[2], Sashank Reddi[3], Sanjiv Kumar[3], Cho-Jui Hsieh[2,3]**
[1]Department of Infomation Science and Electrical Engineering, Zhejiang University
[2]Department of Computer Science, UCLA
[3]Google Research
`ruanyj3107@zju.edu.cn, yhxiong@cs.ucla.edu,`
`{sashank, sanjivk}@google.com, chohsieh@cs.ucla.edu`

## Abstract

In the learning to learn (L2L) framework, we cast the design of optimization algorithms as a machine learning problem and use deep neural networks to learn the update rules. In this paper, we extend the L2L framework to zeroth-order (ZO) optimization setting, where no explicit gradient information is available. Our learned optimizer, modeled as recurrent neural networks (RNNs), first approximates gradient by ZO gradient estimator and then produces parameter update utilizing the knowledge of previous iterations. To reduce the high variance effect due to ZO gradient estimator, we further introduce another RNN to learn the Gaussian sampling rule and dynamically guide the query direction sampling. Our learned optimizer outperforms hand-designed algorithms in terms of convergence rate and final solution on both synthetic and practical ZO optimization problems (in particular, the black-box adversarial attack task, which is one of the most widely used applications of ZO optimization). We finally conduct extensive analytical experiments to demonstrate the effectiveness of our proposed optimizer.[1]

## 1 Introduction

Learning to learn (L2L) is a recently proposed meta-learning framework where we leverage deep neural networks to learn optimization algorithms automatically. The most common choice for the learned optimizer is recurrent neural network (RNN) since it can capture long-term dependencies and propose parameter updates based on knowledge of previous iterations. By training RNN optimizers on predefined optimization problems, the optimizers are capable of learning to explore the loss landscape and adaptively choose descent directions and steps (Lv et al., 2017). Recent works (Andrychowicz et al., 2016; Wichrowska et al., 2017; Lv et al., 2017) have shown promising results that these learned optimizers can often outperform widely used hand-designed algorithms such as SGD, RMSProp, ADAM, etc. Despite great prospects in this field, almost all previous learned optimizers are gradient-based, which cannot be applied to solve optimization problems where explicit gradients are difficult or infeasible to obtain.

Such problems mentioned above are called zeroth-order (ZO) optimization problems, where the optimizer is only provided with function values (zeroth-order information) rather than explicit gradients (first-order information). They are attracting increasing attention for solving ML problems in the black-box setting or when computing gradients is too expensive (Liu et al., 2018a). Recently, one of the most important applications of ZO optimization is the black-box adversarial attack to well-trained deep neural networks, since in practice only input-output correspondence of targeted models rather than internal model information is accessible (Papernot et al., 2017; Chen et al., 2017a).

Although ZO optimization is popular for solving ML problems, the performance of existing algorithms is barely satisfactory. The basic idea behind these algorithms is to approximate gradients via ZO oracle (Nesterov & Spokoiny, 2017; Ghadimi & Lan, 2013). Given the loss function $f$ with its parameter $\boldsymbol{\theta}$ to be optimized (called the *optimizee*), we can obtain its ZO gradient estimator by:

$$\hat{\nabla} f(\boldsymbol{\theta}) = (1/\mu q) \sum_{i=1}^{q} [f(\boldsymbol{\theta} + \mu \boldsymbol{u}_i) - f(\boldsymbol{\theta})] \boldsymbol{u}_i \qquad (1)$$

---

[1]Our code is available at `https://github.com/RYoungJ/ZO-L2L`

where $\mu$ is the smoothing parameter, $\{\boldsymbol{u}_i\}$ are random query directions drawn from standard Gaussian distribution (Nesterov & Spokoiny, 2017) and $q$ is the number of sampled query directions. However, the high variance of ZO gradient estimator which results from both random query directions and random samples (in stochastic setting) hampers the convergence rate of current ZO algorithms. Typically, as problem dimension $d$ increases, these ZO algorithms suffer from increasing iteration complexity by a small polynomial of $d$ to explore the higher dimensional query space.

In this paper, we propose to learn a zeroth-order optimizer. Instead of designing variance reduced and faster converging algorithms by hand as in Liu et al. (2018a;b), we replace parameter update rule as well as guided sampling rule for query directions with learned recurrent neural networks (RNNs). The main contributions of this paper are summarized as follows:

- We extend the L2L framework to ZO optimization setting and propose to use RNN to learn ZO update rules automatically. Our learned optimizer contributes to faster convergence and lower final loss compared with hand-designed ZO algorithms.

- Instead of using standard Gaussian sampling for random query directions as in traditional ZO algorithms, we propose to learn the Gaussian sampling rule and adaptively modify the search distribution. We use another RNN to adapt the variance of random Gaussian sampling. This new technique helps the optimizer to automatically sample on a more important search space and thus results in a more accurate gradient estimator at each iteration.

- Our learned optimizer leads to significant improvement on some ZO optimization tasks (especially the black-box adversarial attack task). We also conduct extensive experiments to analyze the effectiveness of our learned optimizer.

## 2 RELATED WORK

**Learning to learn (L2L)** In the L2L framework, the design of optimization algorithms is cast as a learning problem and deep neural network is used to learn the update rule automatically. In Cotter & Conwell (1990), early attempts were made to model adaptive learning algorithms as recurrent neural network (RNN) and were further developed in Younger et al. (2001) where RNN was trained to optimize simple convex functions. Recently, Andrychowicz et al. (2016) proposed a coordinatewise LSTM optimizer model to learn the parameter update rule tailored to a particular class of optimization problems and showed the learned optimizer could be applied to train deep neural networks. In Wichrowska et al. (2017) and Lv et al. (2017), several elaborate designs were proposed to improve the generalization and scalability of learned optimizers. Li & Malik (2016) and Li & Malik (2017) took a reinforcement learning (RL) perspective and used policy search to learn the optimization algorithms (viewed as RL policies). However, most previous learned optimizers rely on first-order information and use explicit gradients to produce parameter updates, which is not applicable when explicit gradients are not available.

In this paper, we aim to learn an optimizer for ZO optimization problems. The most relevant work to ours is Chen et al. (2017b). In this work, the authors proposed to learn a global black-box (zeroth-order) optimizer which takes as inputs current query point and function value and outputs the next query point. Although the learned optimizer achieves comparable performance with traditional Bayesian optimization algorithms on some black-box optimization tasks, it has several crucial drawbacks. As is pointed out in their paper, the learned optimizer scales poorly with long training steps and is specialized to a fixed problem dimension. Furthermore, it is not suitable for solving black-box optimization problems of high dimensions.

**Zeroth-order (ZO) optimization** The most common method of ZO optimization is to approximate gradient by ZO gradient estimator. Existing ZO optimization algorithms include ZO-SGD (Ghadimi & Lan, 2013), ZO-SCD (Lian et al., 2016), ZO-signSGD (Liu et al., 2019), ZO-ADAM (Chen et al., 2017a), etc. These algorithms suffer from high variance of ZO gradient estimator and typically increase the iteration complexity of their first-order counterparts by a small-degree polynomial of problem dimension $d$. To tackle this problem, several variance reduced and faster converging algorithms were proposed. ZO-SVRG (Liu et al., 2018b) reduced the variance of random samples by dividing optimization steps into several epochs and maintaining a snapshot point at each epoch whose gradient was estimated using a larger or the full batch. And the snapshot point served as a reference in building a modified stochastic gradient estimate at each inner iteration. ZO-SZVR-G

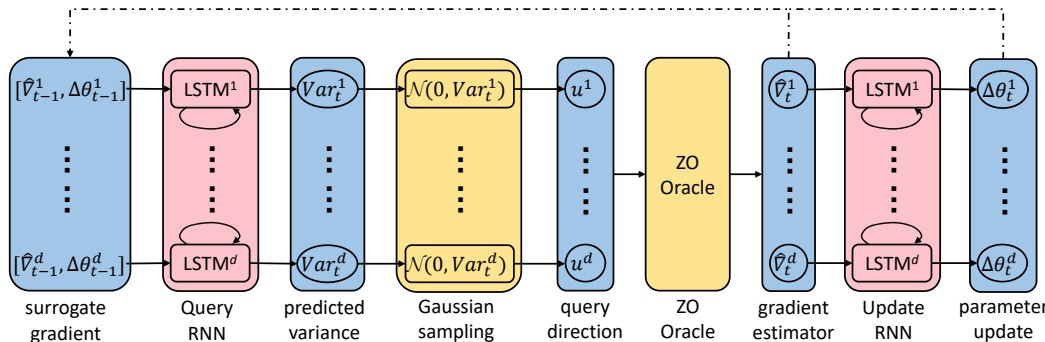

Figure 1: Model architecture of our proposed optimizer. All the operations are applied coordinate-wisely except querying ZO Oracle to obtain ZO gradient estimator (equation 1). Each coordinate shares the QueryRNN and the UpdateRNN parameters but maintains its own hidden state.

(Liu et al., 2018a) adopted a similar strategy and extended to reduce the variance of both random samples and random query directions. But these methods reduce the variance at the cost of higher query complexity. In this paper, we avoid laborious hand design of these algorithms and aim to learn ZO optimization algorithms automatically.

## 3 METHOD

### 3.1 MODEL ARCHITECTURE

Our proposed RNN optimizer consists of three main parts: UpdateRNN, Guided ZO Oracle, and QueryRNN, as shown in Figure 1.

**UpdateRNN** The function of the UpdateRNN is to *learn the parameter update rule* of ZO optimization. Following the idea in Andrychowicz et al. (2016), we use coordinatewise LSTM to model the UpdateRNN. Each coordinate of the optimizee shares the same network but maintains its own separate hidden state, which means that different parameters are optimized using the same update rule based on their own knowledge of previous iterations. Different from previous design in the first-order setting, UpdateRNN takes as input ZO gradient estimator in equation 1 rather than exact gradient and outputs parameter update for each coordinate. Thus the parameter update rule is:

$$\boldsymbol{\theta}_t = \boldsymbol{\theta}_{t-1} + \text{UpdateRNN}(\hat{\nabla} f(\boldsymbol{\theta}_t)) \tag{2}$$

where $\boldsymbol{\theta}_t$ is the optimizee parameter at iteration $t$. Besides learning to adaptively compute parameter updates by exploring the loss landscape, the UpdateRNN can also reduce negative effects caused by the high variance of ZO gradient estimator due to its long-term dependency.

**Guided ZO Oracle** In current ZO optimization approaches, ZO gradient estimator is computed by finite difference along the query direction which is randomly sampled from multivariate standard Gaussian distribution. But this estimate suffers from high variance and leads to poor convergence rate when applied to optimize problems of high dimensions (Duchi et al., 2015). To tackle this problem, we propose to use some prior knowledge learned from previous iterates during optimization to guide the random query direction search and adaptively modify the search distribution. Specifically, at iteration $t$, we use $\mathcal{N}(\mathbf{0}, \boldsymbol{\Sigma}_t)$ to sample query directions ($\boldsymbol{\Sigma}_t$ is produced by the QueryRNN which is introduced later) and then obtain ZO gradient estimator along sampled directions via ZO Oracle (equation 1). The learned adaptive sampling strategy will automatically identify important sampling space which leads to a more accurate gradient estimator under a fixed query budget, thus further increases the convergence rate in ZO optimization tasks. For example, in the black-box adversarial attack task, there is usually a clear important subspace for successful attack, and sampling directions from that subspace will lead to much faster convergence. This idea is similar to that of search distribution augmentation techniques for evolutionary strategies (ES) such as Natural ES (Wierstra et al., 2008), CMA-ES (Hansen, 2016) and Guided ES (Maheswaranathan et al., 2018). However, these methods explicitly define the sampling rule whereas we propose to learn the Gaussian sampling rule (i.e., the adaption of covariance matrix $\boldsymbol{\Sigma}_t$) in an automatic manner.

**QueryRNN** We propose to use another LSTM network called QueryRNN to *learn the Gaussian sampling rule* and dynamically predict the covariance matrix $\mathbf{\Sigma}_t$. We assume $\mathbf{\Sigma}_t$ is diagonal so that it could be predicted in a coordinatewise manner as in the UpdateRNN and the learned QueryRNN is invariant to the dimension of optimizee parameter. It takes as input ZO gradient estimator and parameter update at last iterate (which could be viewed as surrogate gradient information) and outputs the sampling variance coordinatewisely, which can be written as:

$$\mathbf{\Sigma}_t = \text{QueryRNN}([\hat{\nabla} f(\boldsymbol{\theta}_{t-1}), \Delta \boldsymbol{\theta}_{t-1}]) \tag{3}$$

The intuition is that if estimated gradients or parameter updates of previous iterates are biased toward a certain direction, then we can probably increase the sampling probability toward that direction.

Using predicted covariance $\mathbf{\Sigma}_t$ to sample query directions increases the bias of estimated gradient and reduces the variance, which leads to a *tradeoff between bias and variance*. The reduction of variance contributes to faster convergence, but too much bias tends to make the learned optimizer stuck at bad local optima (See more illustrations in Appendix C.3). To balance the bias and variance, at test time we randomly use the covariance of standard Gaussian distribution $\boldsymbol{I}_d$ and the predicted covariance $\mathbf{\Sigma}_t$ as the input of Guided ZO Oracle: $\mathbf{\Sigma}_t' = \text{X}\mathbf{\Sigma}_t + (1 - \text{X})\boldsymbol{I}_d$ where $\text{X} \sim Ber(p)$ is a Bernoulli random variable that trades off between bias and variance. Note that the norm of the sampling covariance $\|\mathbf{\Sigma}_t'\|$ may not equal to that of standard Gaussian sampling covariance $\|\boldsymbol{I}_d\|$, which makes the expectation of the sampled query direction norm $\|\boldsymbol{u}\|$ change. To keep the norm of query direction invariant, we then normalize the norm of $\mathbf{\Sigma}_t'$ to the norm of $\boldsymbol{I}_d$.

### 3.2 OBJECTIVE FUNCTION

The objective function of training our proposed optimizer can be written as follows:

$$\mathcal{L}(\boldsymbol{\phi}) = \sum_{t=1}^{T} \omega_t f(\boldsymbol{\theta}_t(\boldsymbol{\phi})) + \lambda \|\mathbf{\Sigma}_t(\boldsymbol{\phi}) - \boldsymbol{I}_d\|^2 \tag{4}$$

where $\boldsymbol{\phi}$ is the parameter of the optimizer including both the UpdateRNN and the QueryRNN, $\boldsymbol{\theta}_t$ is updated by the optimizer (equation 2) and thus determined by $\boldsymbol{\phi}$, $T$ is the horizon of the optimization trajectory and $\{\omega_t\}$ are predefined weights associated with each time step $t$. The objective function consists of two terms. The first one is the weighted sum of the optimizee loss values at each time step. We use linearly increasing weight (i.e., $\omega_t = t$) to force the learned optimizer to attach greater importance to the final loss rather than focus on the initial optimization stage. The second one is the regularization term of predicted Gaussian sampling covariance $\mathbf{\Sigma}_t$ with regularization parameter $\lambda$. This term prevents the QueryRNN from predicting too big or too small variance value.

### 3.3 TRAINING THE LEARNED OPTIMIZER

In experiments, we do not train the UpdateRNN and the QueryRNN jointly for the sake of stability. Instead, we first train the UpdateRNN using standard Gaussian random vectors as query directions. Then we freeze the parameters of the UpdateRNN and train the QueryRNN separately. Both two modules are trained by truncated Backpropagation Through Time (BPTT) and using the same objective function in equation 4.

In order to backpropagate through the random Gaussian sampling module to train the QueryRNN, we use the reparameterization trick (Kingma & Welling, 2013) to generate random query directions. Specifically, to generate query direction $\boldsymbol{u} \sim \mathcal{N}(\boldsymbol{0}, \mathbf{\Sigma}_t)$, we first sample standard Gaussian vector $\boldsymbol{z} \sim \mathcal{N}(\boldsymbol{0}, \boldsymbol{I}_d)$, and then apply the reparameterization $\boldsymbol{u} = \mathbf{\Sigma}_t^{1/2} \boldsymbol{z}$.

To train the optimizer, we first need to take derivatives of the optimizee loss function w.r.t. the optimizee parameters, and then backpropagate to the optimizer parameters by unrolling the optimziation steps. However, the gradient information of the optimizee is not available in zeroth-order setting. In order to obtain the derivatives, we can follow the assumption in Chen et al. (2017b) that we could get the gradient information of the optimizee loss function at training time, and this information is *not needed at test time*. However, this assumption cannot hold when the gradient of optimizee loss function is not available neither at training time. In this situation, we propose to approximate the gradient of the optimizee loss function w.r.t its parameter using coordinatewise ZO gradient estimator (Lian

et al., 2016; Liu et al., 2018b):

$$\hat{\nabla}f(\boldsymbol{\theta}) = \sum_{i=1}^{d}(1/2\mu_i)[f(\boldsymbol{\theta} + \mu_i \boldsymbol{e}_i) - f(\boldsymbol{\theta} - \mu_i \boldsymbol{e}_i)]\boldsymbol{e}_i \qquad (5)$$

where $d$ is the dimension of the optimizee loss function, $\mu_i$ is the smoothing parameter for the $i^{\text{th}}$ coordinate, and $\boldsymbol{e}_i \in \mathbb{R}^d$ is the standard basis vector with its $i^{\text{th}}$ coordinate being 1, and others being 0s. This estimator is deterministic and could achieve an accurate estimate when $\{\mu_i\}$ are sufficiently small. And it is used only to backpropagate the error signal from the optimizee loss function to its parameter to train the optimizer, which is different from the estimator in equation 1 that is used by the optimizer to propose parameter update. Note that this estimator requires function queries scaled with $d$, which would slow down the training speed especially when optimizee is of high dimension. However, we can compute the gradient estimator of each coordinate in parallel to significantly reduce the computation overhead.

# 4 EXPERIMENTAL RESULTS

In this section, we empirically demonstrate the superiority of our proposed ZO optimizer on a practical application (black-box adversarial attack on MINST and CIFAR-10 dataset) and a synthetic problem (binary classification in stochastic zeroth-order setting). We compare our learned optimizer (called ZO-LSTM below) with existing hand-designed ZO optimization algorithms, including ZO-SGD (Ghadimi & Lan, 2013), ZO-signSGD (Liu et al., 2019), and ZO-ADAM (Chen et al., 2017a). For each task, we tune the hyperparameters of baseline algorithms to report the best performance. Specifically, we set the learning rate of baseline algorithms to $\delta/d$. We first coarsely tune the constant $\delta$ on a logarithmic range $\{0.01, 0.1, 1, 10, 100, 1000\}$ and then finetune it on a linear range. For ZO-ADAM, we tune $\beta_1$ values over $\{0.9, 0.99\}$ and $\beta_2$ values over $\{0.99, 0.996, 0.999\}$. To ensure fair comparison, all optimizers are using the same number of query directions to obtain ZO gradient estimator at each iteration.

In all experiments, we use 1-layer LSTM with 10 hidden units for both the UpdateRNN and the QueryRNN. For each RNN, another linear layer is applied to project the hidden state to the output (1-dim parameter update for the UpdateRNN and 1-dim predicted variance for the QueryRNN). The regularization parameter $\lambda$ in the training objective function (equation 4) is set to 0.005. We use ADAM to train our proposed optimizer with truncated BPTT, each optimization is run for 200 steps and unrolled for 20 steps unless specified otherwise. At test time, we set the Bernoulli random variable (see Section 3.1) $X \sim Ber(0.5)$.

## 4.1 ADVERSARIAL ATTACK TO BLACK-BOX MODELS

We first consider an important application of our learned ZO optimizer: generating adversarial examples to attack black-box models. In this problem, given the targeted model $F(\cdot)$ and an original example $\boldsymbol{x}_0$, the goal is to find an adversarial example $\boldsymbol{x}$ with small perturbation that minimizes a loss function $\text{Loss}(\cdot)$ which reflects attack successfulness. The black-box attack loss function can be formulated as $f(\boldsymbol{x}) = c\|\boldsymbol{x} - \boldsymbol{x}_0\| + \text{Loss}(F(\boldsymbol{x}))$, where $c$ balances the perturbation norm and attack successfulness (Carlini & Wagner, 2017; Tu et al., 2019). Due to the black-box setting, one can only compute the function value of the above objective, which leads to ZO optimization problems (Chen et al., 2017a). Note that attacking each sample $\boldsymbol{x}_0$ in the dataset corresponds to a particular ZO optimization problem instance, which motivates us to train a ZO optimizer (or "attacker") offline with a small subset and apply it to online attack to other samples with faster convergence (which means lower query complexity) and lower final loss (which means less distortion).

Here we experiment with black-box attack to deep neural network image classifier, see detailed problem formulation in Appendix A.1. We follow the same neural network architectures used in Cheng et al. (2019) for MNIST and CIFAR-10 dataset, which achieve 99.2% and 82.7% test accuracy respectively. We randomly select 100 images that are correctly classified by the targeted model in each test set to train the optimizer and select another 100 images to test the learned optimizer. The dimension of the optimizee function is $d = 28 \times 28$ for MNIST and $d = 32 \times 32 \times 3$ for CIFAR-10. The number of sampled query directions is set to $q = 20$ for MNIST and $q = 50$ for CIFAR-10 respectively. All optimizers use the same initial points for finding adversarial examples.

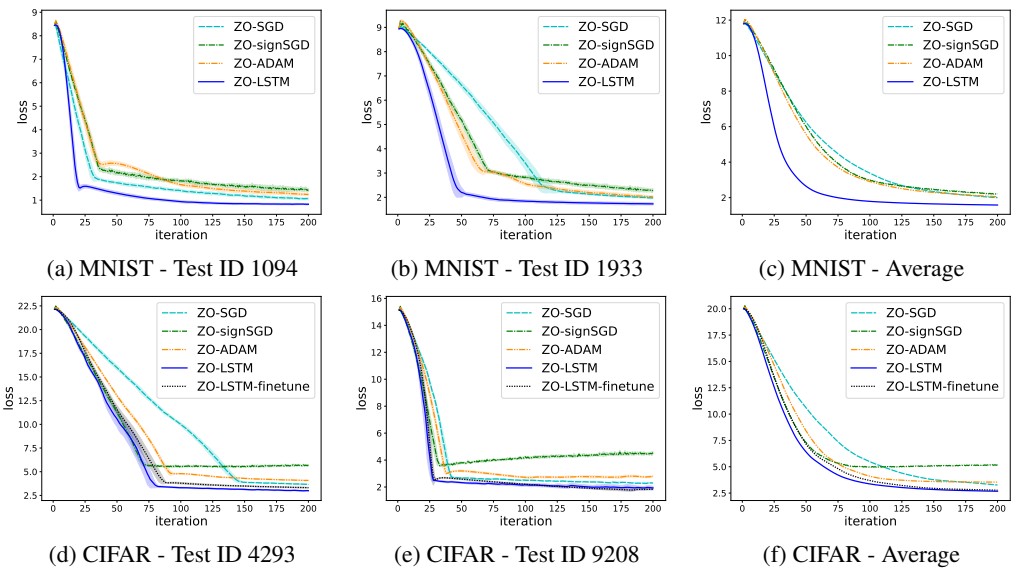

Figure 2: (a)-(b) & (d)-(e): Black-box attack loss versus iterations on selected test images. The loss curves are averaged over 10 independent random trails and the shaded areas indicate the standard deviation. (c) & (f): Black-box attack loss curves averaged over all 100 test images. Attack on each image is run for 10 trails. On CIFAR-10 attack task, we also test the performance of the learned optimizer trained on MINST attack task with finetuning (ZO-LSTM-finetune, which will be introduced in Section 4.3).

Figure 2 shows black-box attack loss versus iterations using different optimizers. We plot the loss curves of two selected test images (see Appendix A.3 for more plots on other test images) as well as the average loss curve over all 100 test images for each dataset. It is clear that our learned optimizer (ZO-LSTM) leads to much faster convergence and lower final loss than other baseline optimizers both on MNIST and CIFAR-10 attack tasks. The visualization of generated adversarial examples versus iterations can be found in Appendix A.2.

## 4.2 STOCHASTIC ZEROTH-ORDER BINARY CLASSIFICATION

Next we apply our learned optimizer in the stochastic ZO optimization setting. We consider a synthetic binary classification problem (Liu et al., 2019) with non-convex least squared loss function: $f(\boldsymbol{\theta}) = \frac{1}{n} \sum_{i=1}^{n} (y_i - 1/(1 + e^{-\boldsymbol{\theta}^T \boldsymbol{x}_i}))^2$. To generate one dataset for the binary classification task, we first randomly sample a $d$-dimensional vector $\boldsymbol{\theta} \in \mathbb{R}^d$ from $\mathcal{N}(\mathbf{0}, \boldsymbol{I}_d)$ as the ground-truth. Then we draw samples $\{\boldsymbol{x}_i\}$ from $\mathcal{N}(\mathbf{0}, \boldsymbol{I}_d)$ and obtain the label $y_i = 1$ if $\boldsymbol{\theta}^T \boldsymbol{x}_i > 0$ and $y_i = 0$ otherwise. The size of training set is 2000 for each dataset. Note that each dataset corresponds to a different optimizee function in the class of binary classification problem. We generate 100 different datasets in total, and use 90 generated datasets (i.e. 90 binary classification problem instances) to train the optimizer and other 10 to test the performance of the learned optimizer. Unless specified otherwise, the problem dimension is $d = 100$; the batch size and the number of query directions are set to $b = 64$ and $q = 20$ respectively. At each iteration of training, the optimizer is allowed to run 500 steps and unrolled for 20 steps.

In Figure 3a, we compare various ZO optimizers and observe that our learned optimizer outperforms all other hand-designed ZO optimization algorithms. Figure 3b-3c compares the performance of ZO-SGD and ZO-LSTM with different query direction number $q$ and batch size $b$. ZO-LSTM consistently outperforms ZO-SGD in different optimization settings. In Figure 3d, we generate binary classification problems with different dimension $d$ and test the performance of ZO-LSTM. Our learned optimizer generalizes well and still achieves better performance than ZO-SGD.

## 4.3 GENERALIZATION OF THE LEARNED OPTIMIZER

In previous experiments, we train the optimizer using a small subset of problem instances in a particular ZO optimization task and apply the learned optimizer in other problem instances, which

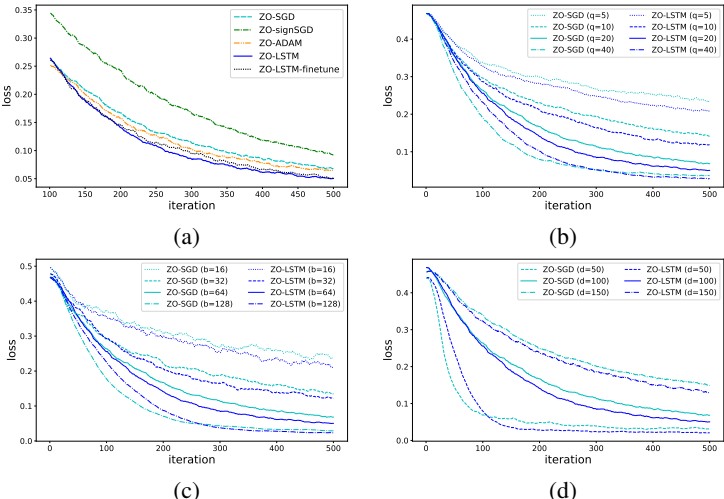

Figure 3: Optimization performance comparison on synthetic binary classification problem. Each line is averaged over 10 test datasets with random initialization. (a): Training loss against iterations. ZO-LSTM-finetune denotes the learned optimizer trained on the MNIST attack task and fintuned on the binary classification task (see Section 4.3). (b)-(d): Effects of query direction number $q$, batch size $b$, and problem dimension $d$, respectively.

demonstrates the generalization in a specific class of ZO optimization problems. In this subsection, we study the generalization of the learned optimizer across different classes of ZO optimization problems.

Current L2L framework (first-order) aims to train an optimizer on a small subset of problems and make the learned optimizer generalize to a wide range of different problems. However, in practice, it is difficult to train a general optimizer that can achieve good performance on problems with different structures and loss landscapes. In experiments, we find that the learned optimizer could not easily generalize to those problems with different relative scales between parameter update and estimated gradient (similar to the definition of learning rate). Therefore, we scale the parameter update produced by the UpdateRNN by a factor $\alpha$ when generalizing the learned optimizer to another totally different task and tune this hyperparameter $\alpha$ on that task (similar to SGD/Adam).

We first train the optimizer on MNIST attack task and then finetune it on CIFAR-10 attack task[2], as shown in Figure 2d-2f. We see that the finetuned optimizer (ZO-LSTM-finetune) achieves comparable performance with ZO-LSTM which is trained from scratch on a CIFAR-10 subset. We also generalize the learned optimizer trained on the MNIST attack task to the totally different binary classification task (Figure 3a) and surprisingly find that it could achieve almost identical performance with ZO-LSTM directly trained on this target task. These results demonstrate that our optimizer has learned a rather general ZO optimization algorithm which can generalize to different classes of ZO optimization problems well.

## 4.4 ANALYSIS

In this section, we conduct experiments to analyze the effectiveness of each module and understand the function mechanism of our proposed optimizer (especially the QueryRNN).

**Ablation study** To assess the effectiveness of each module, we conduct ablation study on each task, as shown in Figure 4a-4c. We compare the performance of ZO-SGD, ZO-LSTM (our model), ZO-LSTM-no-query (our model without the QueryRNN, i.e., use standard Gaussian sampling), ZO-LSTM-no-update (our model without the UpdateRNN, i.e., ZO-SGD with the QueryRNN). We observe that both the QueryRNN and the UpdateRNN improves the performance of the learned

---

[2]Although black-box attack tasks on MNIST and CIFAR-10 dataset seem to be similar on intuition, the ZO optimization problems on these two datasets are not such similar. Because targeted models are of very different architectures and image features also vary a lot, the loss landscape and gradient scale are rather different.

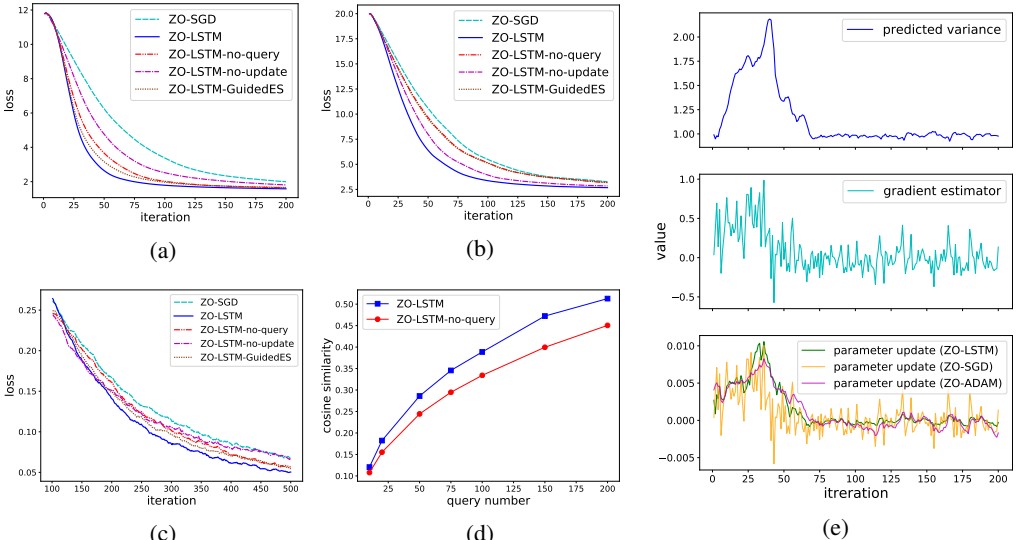

Figure 4: Analytical experiments demonstrating the effectiveness of our proposed optimizer. (a)-(c): Ablation study on MNIST attack task, CIFAR-10 attack task, and binary classification task respectively. For MINST and CIFAR-10 attack task, We average the loss curve over all 100 test images and attack on each image is run for 10 independent trails (see Appendix A.4 for additional plots on single test images). (d): Evaluation of average cosine similarity between ZO gradient estimator and ground-truth gradient with and without the QueryRNN. (e): Optimization trajectory of one coordinate when applying ZO-LSTM on MNIST attack task. In the bottom figure, we apply ZO-SGD and ZO-ADAM to the same optimization trajectory as ZO-LSTM, i.e., assume they use the same ZO gradient estimator at each iteration but produce different parameter updates.

optimizer in terms of convergence rate or/and final solution. Noticeably, the improvement induced by the QueryRNN is less significant on binary classification task than on black-box attack task. We conjecture the reason is that the gradient directions are more random in binary classification task so it is much more difficult for the QueryRNN to identify the important sampling space. To further demonstrate the effectiveness of the QueryRNN, we also compare ZO-LSTM, ZO-LSTM-no-query with ZO-LSTM-GuidedES, whose parameter update is produced by the UpdateRNN but the covariance matrix of random Gaussian sampling is adapted by guided evolutionary strategy (Guided ES). For fair comparison, we use the ZO gradient estimator and the parameter update at last iterate (the same as the input of our QueryRNN) as surrogate gradients for GuidedES (see Appendix B for details). We find that using GuidedES to guide the query direction search also improves the convergence speed on MNIST attack task, but the improvement is much less than that of the QueryRNN. In addition, GuidedES leads to negligible effects on the other two tasks.

**Estimated gradient evaluation** In this experiment, we evaluate the estimated gradient produced by the Guided ZO Oracle with and without the QueryRNN. Specifically, we test our learned optimizer on MNIST attack task and compute the average cosine similarity between the ground-truth gradient and the ZO gradient estimator over optimization steps before convergence. In Figure 4d, we plot the average cosine similarity of ZO-LSTM and ZO-LSTM-no-query against different query direction number $q$. We observe that the cosine similarity becomes higher with the QueryRNN, which means that the direction of ZO gradient estimator is closer to that of the ground-truth gradient. And with the query direction number $q$ increasing, the improvement of cosine similarity becomes more significant. These results can explain the effectiveness of the QueryRNN in terms of obtaining more accurate ZO gradient estimators. In Appendix C.1, we evaluate the iteration complexity with and without the QueryRNN to further verify its improved convergence rate and scalability with problem dimension.

**Optimization trajectory analysis** To obtain a more in-depth understanding of what our proposed optimizer learns, we conduct another analysis on the MNIST attack task. We use the learned optimizer (or "attacker") to attack one test image in the MNIST dataset. Then we select one pixel in the image (corresponds to one coordinate to be optimized), and trace the predicted variance, the gradient estimator and the parameter update of that coordinate at each iteration, as shown in Figure 4e. We

can observe that although the ZO gradient estimator is noisy due to the high variance of random Gaussian sampling, the parameter update produced by the UpdateRNN is less noisy, which makes the optimization process less stochastic. The smoothing effect of the UpdateRNN is similar to that of ZO-ADAM, but it is learned automatically rather than by hand design. The predicted variance produced by the QueryRNN is even smoother. With a larger value of ZO gradient estimator or the parameter update, the QueryRNN produces a larger predicted variance to increase the sampling bias toward that coordinate. We observe that the overall trend of the predicted variance is more similar to that of the parameter update, which probably means the parameter update plays a more important role in the prediction of the Gaussian sample variance. Finally, in Appendix C.2, we also visualize the predicted variance by the QueryRNN and compare it with final added perturbation to the image (i.e., the final solution of attack task).

## 5 CONCLUSION

In this paper, we study the learning to learn framework for zeroth-order optimization problems. We propose a novel RNN-based optimizer that learns both the update rule and the Gaussian sampling rule. Our learned optimizer leads to significant improvement in terms of convergence speed and final loss. Experimental results on both synthetic and practical problems validate the superiority of our learned optimizer over other hand-designed algorithms. We also conduct extensive analytical experiments to show the effectiveness of each module and to understand our learned optimizer.

Despite the prospects of the L2L framework, current learned optimizers still have several drawbacks, such as lack of theoretical convergence proof and extra training overhead. In our future work, we aim to prove the improved convergence in theory and further improve the training methodology.

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

## A  APPLICATION: ADVERSARIAL ATTACK TO BLACK-BOX MODELS

### A.1  PROBLEM FORMULATION FOR BLACK-BOX ATTACK

We consider generating adversarial examples to attack black-box DNN image classifier and formulate it as a zeroth-order optimization problem. The targeted DNN image classifier $F(\boldsymbol{x}) = [F_1, F_2, ..., F_K]$ takes as input an image $\boldsymbol{x} \in [0, 1]^d$ and outputs the prediction scores (i.e. log probabilities) of $K$ classes. Given an image $\boldsymbol{x}_0 \in [0, 1]^d$ and its corresponding true label $t_0 \in [1, 2, .., K]$, an adversarial example $\boldsymbol{x}$ is visually similar to the original image $\boldsymbol{x}_0$ but leads the targeted model $F$ to make wrong prediction other than $t_0$ (i.e., untargeted attack). The black-box attack loss is defined as:

$$\min_{\boldsymbol{x}} \max\{F_{t_0}(\boldsymbol{x}) - \max_{j \neq t_0} F_j(\boldsymbol{x}), 0\} + c\|\boldsymbol{x} - \boldsymbol{x}_0\|_p \tag{6}$$

The first term is the attack loss which measures how successful the adversarial attack is and penalizes correct prediction by the targeted model. The second term is the distortion loss ($p$-norm of added perturbation) which enforces the perturbation added to be small and $c$ is the regularization coefficient. In our experiment, we use $\ell_1$ norm (i.e., $p = 1$), and set $c = 0.1$ for MNIST attack task and $c = 0.25$ for CIFAR-10 attack task. To ensure the perturbed image still lies within the valid image space, we can apply a simple transformation $\boldsymbol{x} = (\tanh(\boldsymbol{w}) + 1)/2$ such that $\boldsymbol{x} \in [0, 1]^d$. Note that in practice, we can only get access to the inputs and outputs of the targeted model, thus we cannot obtain explicit gradients of above loss function, rendering it a ZO optimization problem.

### A.2  VISUALIZATION OF GENERATED ADVERSARIAL EXAMPLES VERSUS ITERATIONS

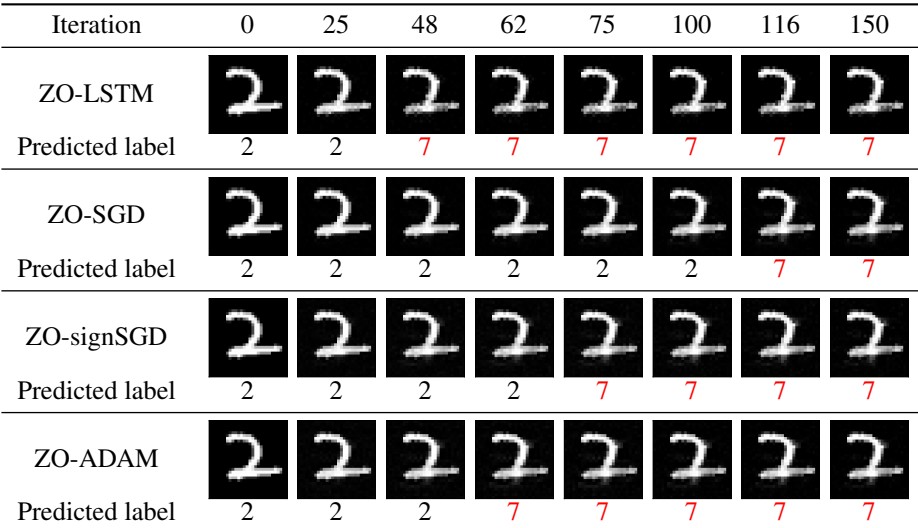

| Iteration | 0 | 25 | 48 | 62 | 75 | 100 | 116 | 150 |
|---|---|---|---|---|---|---|---|---|
| ZO-LSTM | | | | | | | | |
| Predicted label | 2 | 2 | 7 | 7 | 7 | 7 | 7 | 7 |
| ZO-SGD | | | | | | | | |
| Predicted label | 2 | 2 | 2 | 2 | 2 | 2 | 7 | 7 |
| ZO-signSGD | | | | | | | | |
| Predicted label | 2 | 2 | 2 | 2 | 7 | 7 | 7 | 7 |
| ZO-ADAM | | | | | | | | |
| Predicted label | 2 | 2 | 2 | 7 | 7 | 7 | 7 | 7 |

Table 1: Generated adversarial examples of each optimization algorithms versus iterations on MNIST Test ID 1933 (corresponding black-box attack loss curve is shown in Figure 2b).

A.3    ADDITIONAL PLOTS OF BLACK-BOX ATTACK LOSS VERSUS ITERATIONS

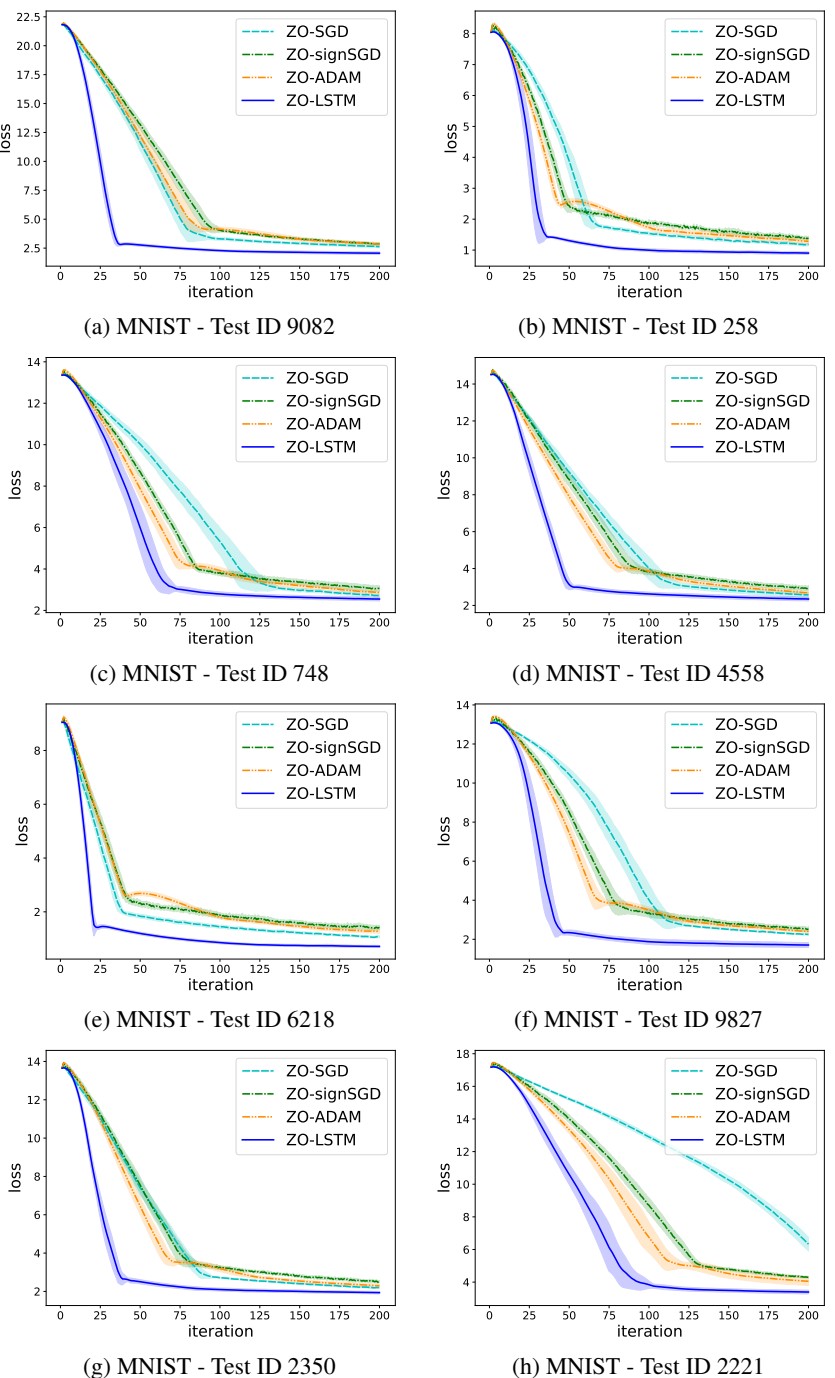

Figure 5: Additional plots of black-box attack loss curves on random selected MNIST test images. The loss curves are averaged over 10 independent random trails and the shaded areas indicate the standard deviation.

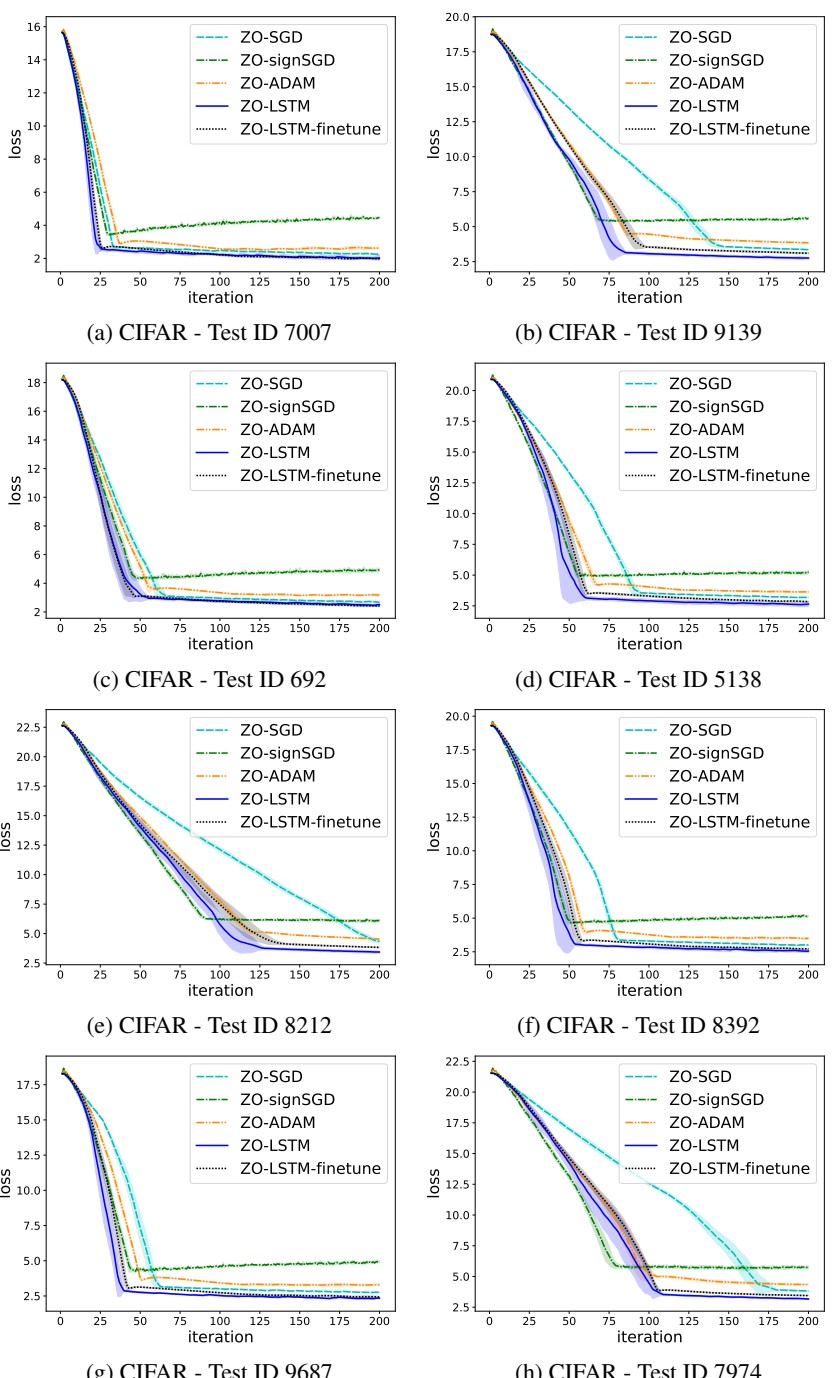

Figure 6: Additional plots of black-box attack loss curves on random selected CIFAR-10 test images. The loss curves are averaged over 10 independent random trails and the shaded areas indicate the standard deviation.

A.4   ADDITIONAL PLOTS FOR ABLATION STUDY

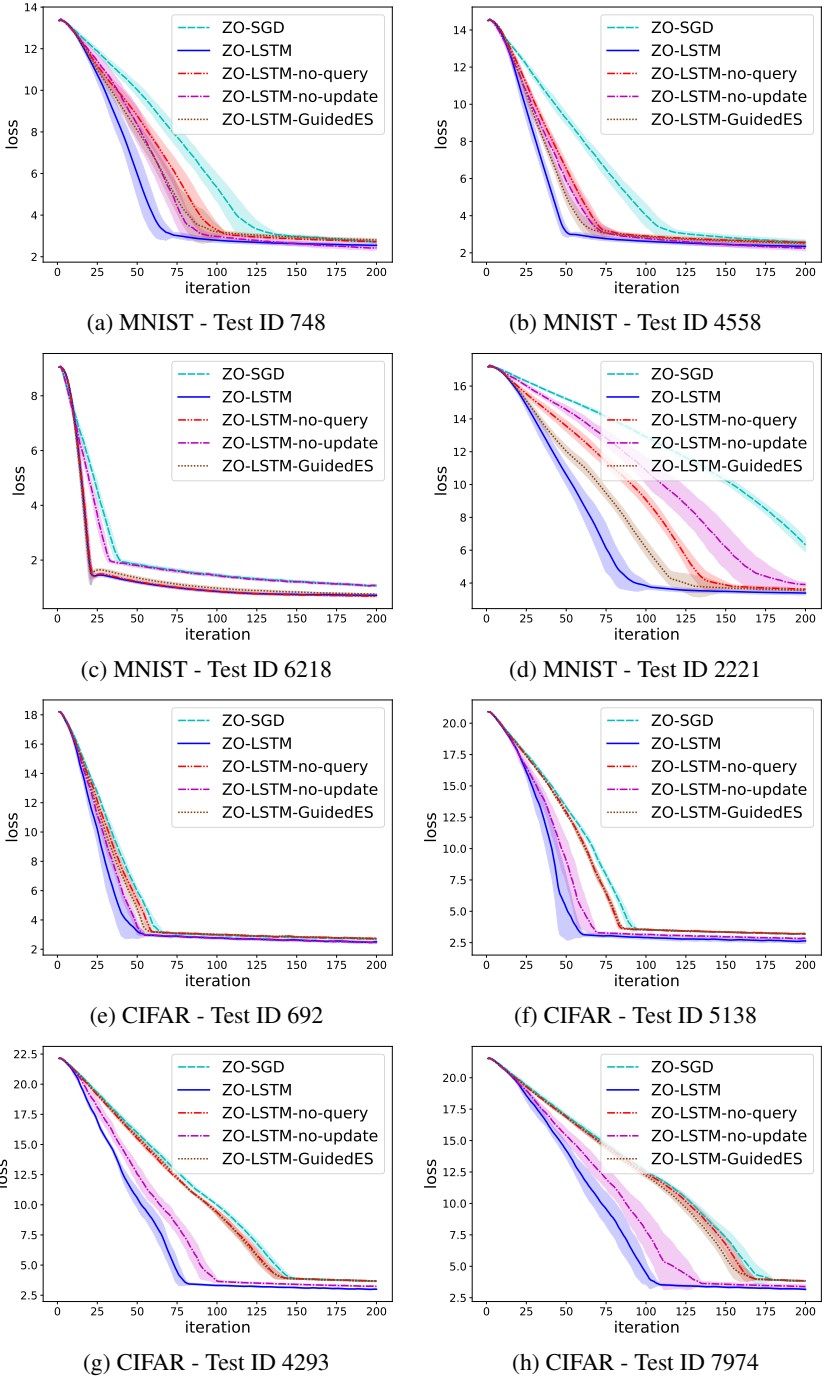

Figure 7: Additional plots for ablation study on single test images. (a)-(d): Plots on randomly selected test images in MNIST dataset. (e)-(h): Plots on randomly selected test images in CIFAR-10 dataset.

## B  IMPLEMENTATION DETAILS FOR GUIDED EVOLUTIONARY STRATEGY

Guided evolutionary strategy (GuidedES) in Maheswaranathan et al. (2018) incorporates surrogate gradient information (which is correlated with true gradient) into random search. It keeps track of a low dimensional guided subspace defined by $k$ surrogate gradients, which is combined with the full space for query direction sampling. Denote $U \in \mathbb{R}^{d \times k}$ as the orthonormal basis of the guided subspace (i.e., $U^T U = I_k$), GuidedES samples query directions from distribution $\mathcal{N}(0, \Sigma)$, where the covariance matrix $\Sigma$ is modified as:

$$\Sigma = \alpha I_d + (1 - \alpha) U U^T \tag{7}$$

where $\alpha$ trades off between the full space and the guided space and we tune the hyperparameter $\alpha = 0.5$ with the best performance in our experiments. Similar to what we have discussed in Section 3.1, we normalize the norm of sampled query direction to keep it invariant. In our experiments, GuidedES uses the ZO gradient estimator and the parameter update at last iterate (the same as the input of our QueryRNN) as input for fair comparison with our proposed QueryRNN.

## C  ADDITIONAL ANALYTICAL STUDY

### C.1  ITERATION COMPLEXITY VERSUS PROBLEM DIMENSION

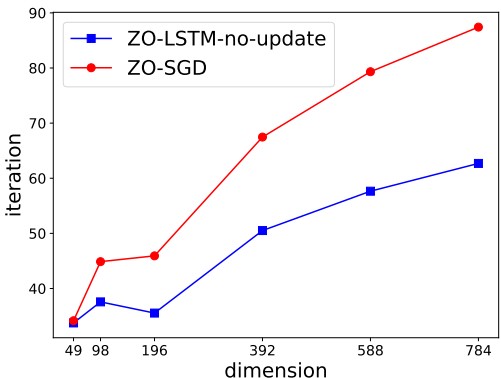

Figure 8: Iteration complexity versus problem dimension on MNIST attack task. Iteration complexity is defined as iterations required to achieve initial attack success which are averaged over 100 test images.

In this experiment, we evaluate the iteration complexity with and without the QueryRNN. Specifically, we test the performance of ZO-SGD and ZO-LSTM-no-update (i.e., ZO-SGD with the QueryRNN) on MNIST attack task and compare the iterations required to achieve initial attack success. In Figure 8, we plot iteration complexity against problem dimension $d$. We generate MNIST attack problems with different dimensions $d \in \{28 \times 28, 21 \times 28, 14 \times 28, 14 \times 14, 7 \times 14, 7 \times 7\}$ by rescaling the added perturbation using bilinear interpolation method. From Figure 8, we find that with the problem dimension increasing, ZO-SGD scales poorly and requires much more iterations (i.e., function queries) to attain initial attack success. With the QueryRNN, ZO-LSTM-no-update consistently requires lower iteration complexity and leads to more significant improvement on problems of higher dimensions. These results show the effectiveness of the QueryRNN in terms of convergence rate and scalability with problem dimensions.

### C.2  VISUALIZATION OF ADDED PERTURBATION AND PREDICTED VARIANCE

To further verify the effectiveness of the QueryRNN, we select one image from MNIST dataset and visualize final added perturbation to the image (i.e., the final solution of MNIST attack task) as well as sampling variance predicted by the QueryRNN, as illustrated in Figure 9. We first compare final perturbation produced by ZO-LSTM (Figure 9b) and ZO-LSTM-no-query (Figure 9c). We observe

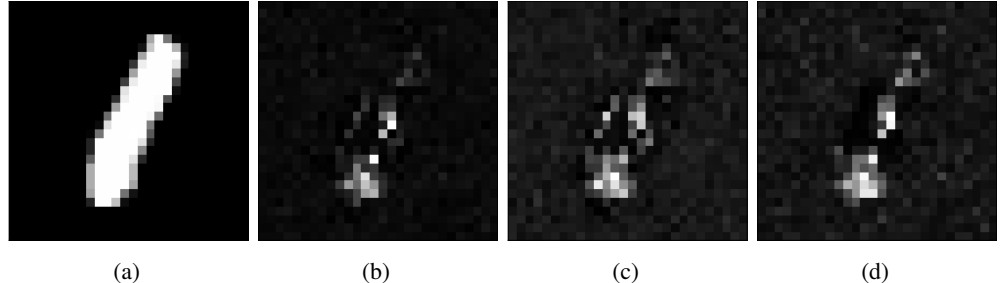

|       |       |       |       |
|-------|-------|-------|-------|
| (a)   | (b)   | (c)   | (d)   |

Figure 9: Visualization of final added perturbation to the image and predicted variance by the QueryRNN. (a): Original image of digit class "1". (b): Final perturbation generated by ZO-LSTM (with the QueryRNN). (c): Final perturbation generated by ZO-LSTM-no-query (without the QueryRNN). (d): Average predicted variance by the QueryRNN of ZO-LSTM over iterations before convergence.

that the perturbation produced by these two optimizers are generally similar, but that produced by ZO-LSTM is less distributed due to the sampling bias induced by the QueryRNN. Then we take the predicted variance by the QueryRNN of ZO-LSTM (averaged over iterations before convergence) into comparison (Figure 9d). We find that there are some similar patterns between average predicted variance by the QueryRNN and final added perturbation generated by ZO-LSTM. It is expected since ZO-LSTM uses the predicted variance by the QueryRNN to sample query directions, which would thus guide the optimization trajectory and influence the final solution. Surprisingly, we see that the average predicted variance by the QueryRNN of ZO-LSTM is also similar to final perturbation produced by ZO-LSTM-no-query (which doesn't utilize the QueryRNN). These results demonstrate that the QueryRNN could recognize useful features quickly in the early optimization stage and produces sampling space toward the final solution.

## C.3  ILLUSTRATION OF THE TRADEOFF BETWEEN BIAS AND VARIANCE

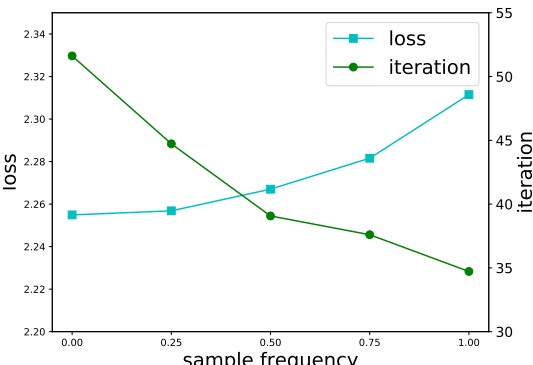

Figure 10: Sensitivity analysis of sample frequency in the predicted subspace on MNIST attack task. Iteration complexity and loss are defined as iterations required to achieve initial attack success and the corresponding loss, which are both averaged over 100 test images.

This experiment means to illustrate the concept of the tradeoff between bias and variance (Section 3.1). We test our learned optimizer on MNIST attack task with different sample frequency in the predicted subspace (i.e., the probability $p$ for the Bernoulli variable $X \sim Ber(p)$ in Section 3.1). As shown in Figure 10, with the sampling frequency increasing, the learned optimizer converges faster but obtains higher loss, which means reduced variance and increased bias respectively. Notably, the sensitivity (i.e., the relative difference) of the final loss w.r.t the sampling frequency is much lower than that of the iteration complexity, which means that we can sample in the predicted

subspace with a higher frequency. In our experiments, we simply set the sampling frequency to 0.5 without extra tuning.

## C.4 COMPARISON WITH VARIANCE REDUCED ALGORITHM

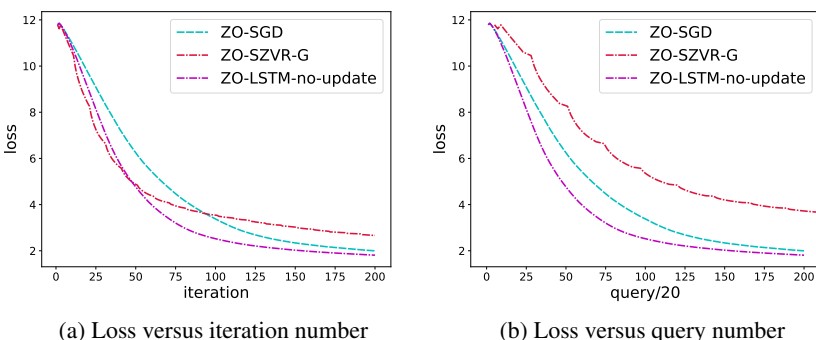

(a) Loss versus iteration number      (b) Loss versus query number

Figure 11: Comparison between with existing zeroth-order variance reduced algorithm (ZO-SZVR-G) on MNIST attack task. Loss curves are averaged over all 100 test images and attack on each image is run for 10 trails.

In this experiment, we compare the performance of ZO-SGD with the QueryRNN (ZO-LSTM-no-update) and ZO-SGD with the variance reduced method (ZO-SZVR-G) on MNIST attack task. For fair comparison, each method uses $q = 20$ query directions to obtain ZO gradient estimator at each iteration. For ZO-SZVR-G, we divide iterations into epochs of length 10. At the beginning of each epoch, we maintain a snapshot whose gradient is estimated using $q' = 100$ query directions and this snapshot is used as a reference to modify the gradient estimator at each inner iteration. We refer readers to Liu et al. (2018a) for more details.

In Figure 11a, we compare the black-box attack loss versus iterations. We observe that although ZO-SZVR-G converges faster than ZO-SGD because of reduced variance, it leads to higher final loss values. But our QueryRNN brings about improvements both in terms of convergence rate and final loss. Note that ZO-SZVR-G requires more function queries to obtain the snapshot and modify the gradient estimator at each iteration, we also plot black-box attack loss versus queries in Figure 11b. We observe that ZO-SZVR-G needs much more queries than ZO-SGD and our method.

