# OpenReview forum: "Learning to Learn by Zeroth-Order Oracle"
_ICLR.cc/2020/Conference — Accept (Poster)_

### Official Review · AnonReviewer3 · 2019-10-23
**Official Blind Review #3**

**Rating:** 6

**Review:**

The paper proposes a zeroth-order optimization framework that employs an RNN to modulate the sampling used to estimate gradients and a second RNN that models the parameter update. More specifically, query directions are sampled from a Gaussian distribution with a diagonal covariance whose evolution is determined by an RNN (QueryRNN). The resulting gradient estimates are used by an RNN (UpdateRNN) that learns the parameter update rule. This framework has the stated advantage that, unlike existing work in ZO optimization, it does not rely upon hand-designed strategies to reduce the variance common to ZO gradient estimators. The paper evaluates the proposed framework on MNIST and CFAR tasks, as well as a synthetic binary classification task. The results demonstrate faster convergence compared to baseline zeroth-order optimization algorithms, while ablations indicate the contributions of the different model components.

The primary contribution of the proposed method is the inclusion of a second network that learns to adapt the covariance of the Gaussian from which  query directions are sampled. The use of an RNN to model the parameter update rule is borrowed from previous work. The ablation studies show that the adaptive sampling strategy noticeably improves convergence as does the inclusion of the RNN update (the contribution of UpdateRNN is more significant in one ablation study, while the contribution of QueryRNN is greater in the other).

Given that a stated advantage of QueryRNN is reducing variance, it would be beneficial to compare against baselines such as Liu et al., 2018a,b and/or Guo et al., 2016 which similarly seek to reduce variance.

The paper includes a large number of typos (e.g., CIFAR-->CIAFR) and grammatical errors, but is otherwise clear.

ADDITIONAL COMMENTS/QUESTIONS

* The related work discussion would benefit from a discussion of how Liu et al. 2018 and Guo et al. 2016 reduce variance

* The computational complexity of the proposed method as compared to the baselines is unclear as is the scalability with dimensionality. At various points, the paper comments that other methods scale poorly with the dimensionality of the query space, which is true of the proposed method unless the operations are parallelized. Is this not possible with the baseline methods?

* The paper makes hand wavy claims to the fact that modulating the diagonal covariance matrix allows the method to focus on certain subspaces. It would be helpful to make these claims more formal, particularly in light of the fact that the mean does not change.

* The method relies upon a bit of a hack that samples from a standard Gaussian at random times. How important is this to performance? How sensitive is convergence to the frequency with which standard Gaussian sampling is used?

* The discussion of Eqn. 5 as it relates to Eqn. 1 is unclear.


UPDATE AFTER AUTHOR RESPONSE

I appreciate the authors' thorough response, which resolved my primary questions/concerns, including comparisons to existing variance reduction methods (which should be incorporated into the main paper).

**Experience Assessment:**

I do not know much about this area.

**Review Assessment: Checking Correctness Of Derivations And Theory:**

I assessed the sensibility of the derivations and theory.

**Review Assessment: Checking Correctness Of Experiments:**

I carefully checked the experiments.

**Review Assessment: Thoroughness In Paper Reading:**

I read the paper thoroughly.

---

> ### Author Response · Authors · 2019-11-15
> **Response to Reviewer #3 (2/2)**
>
> Thank you for your comments and suggestions! Here are our responses to your questions.
>
> Q5: The paper makes hand wavy claims to the fact that modulating the diagonal covariance matrix allows the method to focus on certain subspaces. It would be helpful to make these claims more formal, particularly in light of the fact that the mean does not change.
>
> As for why modulating the diagonal covariance matrix allows our method to focus on certain subspace, we can simply consider the two-dimensional optimization problem. By using standard Gaussian distribution, we sample query directions from a circle and the probability distribution is isotropy. But by using the predicted diagonal covariance, we sample query directions from an ellipse and the probability distribution will be biased toward a certain axis, which means we increase the sampling probability in a certain subspace instead of uniformly sampling in the full space.
>
> As for the sampling mean, in fact, we have experimented with also predicting the sampling mean but it tends to make the learned optimizer easily stuck at local optima. Thus, we just fix the zero mean and predict the sampling covariance.
>
>
> Q6: The method relies upon a bit of a hack that samples from a standard Gaussian at random times. How important is this to performance? How sensitive is convergence to the frequency with which standard Gaussian sampling is used?
>
> We have conducted another experiment (added to Appendix C.3) on the MNIST attack task to analyze the sensitivity of the final loss and iteration complexity w.r.t to the sampling frequency. As the sampling frequency varies from 0 to 1, we observe that the number of iterations drops significantly while the converged loss is relatively stable (from 2.25 to 2.31). This suggests the algorithm is not very sensitive to this parameter and in practice we can set the sampling frequency to a larger number. In our experiments, we simply set the sampling frequency to 0.5 without extra tuning. Please refer to Appendix C.3 for more details.
>
>
> Q7: The discussion of Eqn. 5 as it relates to Eqn. 1 is unclear.
>
> Equation (1) and equation (5) are both widely used zeroth-order gradient estimators but play different roles in our proposed method. The gradient estimator in equation (1) serves as the input of our UpdateRNN to propose the parameter update. To reduce query complexity, we usually only sample a small number of query directions and approximate gradients along sampled query directions. But the function of equation (5) is completely different. Since our objective function (equation (4)) of training the optimizer contains the loss function of the optimizee whose gradients are not available in zeroth-order setting, we cannot backpropagate through it directly. To make training feasible, we use the gradient estimator in equation (5) to approximate the gradient of the optimizee. This estimator is deterministic and could achieve an accurate estimate to stabilize training if the smoothing parameters {\mu_i} are sufficiently small, but requires query complexity scaled with problem dimension. Thus, we apply parallelization to compute equation (5) to reduce the computation overhead.

---

> ### Author Response · Authors · 2019-11-15
> **Response to Reviewer #3 (1/2)**
>
> Thank you for your comments and suggestions! Here are our responses to your questions.
>
> Your first and third questions are both about existing variance reduced methods for zeroth-order optimization. We make response to these two questions together.
>
> Q1: Given that a stated advantage of QueryRNN is reducing variance, it would be beneficial to compare against baselines such as Liu et al., 2018a,b and/or Guo et al., 2016 which similarly seek to reduce variance.
>
> Q3: The related work discussion would benefit from a discussion of how Liu et al. 2018 and Guo et al. 2016 reduce variance
>
> About related work: ZO-SVRG (Liu et al.,2018b) reduced the variance of random samples by dividing optimization steps into several epochs and maintaining a snapshot point at each epoch whose gradient was estimated using a larger or the full batch. And the snapshot point served as a reference in building a modified stochastic gradient estimate at each inner iteration. ZO-SZVR-G (Liu et al.,2018a) adopted a similar strategy and extended it to reduce the variance of both random samples and random query directions. AsyDSZOVR (Gu et al., 2016) applied a similar variance reduction method in the asynchronous zeroth-order optimization setting. We have added this discussion to the related work part.
>
> Comparison with existing methods: Since ZO-SZVR-G is the extension of ZO-SVRG and AsyDSZOVR applies in a different optimization setting (asynchronous zeroth-order optimization), we only compare ZO-SZVR-G with our proposed method. We have conducted experiments on MNIST attack task which have been included in Appendix C.4. The main result is that ZO-SZVR-G converges faster than ZO-SGD because of reduced variance but leads to higher final loss values, whereas our QueryRNN brings about improvements both in terms of convergence rate and final loss. Since ZO-SZVR-G requires extra queries to reduce variance at each iteration and each epoch, we also plot the training loss against query number and observe that ZO-SZVR-G needs more queries than ZO-SGD and our method.
>
>
> Q2: The paper includes a large number of typos (e.g., CIFAR-->CIAFR) and grammatical errors, but is otherwise clear.
>
> We are sorry for our mistakes and have corrected them in the revised version.
>
>
> Q4: The computational complexity of the proposed method as compared to the baselines is unclear as is the scalability with dimensionality. At various points, the paper comments that other methods scale poorly with the dimensionality of the query space, which is true of the proposed method unless the operations are parallelized. Is this not possible with the baseline methods?
>
> Sorry for the confusion. For ZO optimizers it’s common to compare the number of queries (or the number of iterations in our experiments because the number of queries is the same at each iteration) required to obtain a certain objective function value, so “scalability” here mainly indicates the number of queries. In all the comparisons, our method can reduce the number of queries over existing optimizers. Also, we plot the number of queries with respect to dimensionality in Appendix C.1 to illustrate the effectiveness of the QueryRNN in terms of scalability with problem dimensions.
>
> For the parallelization of equation (5), it is only for accelerating training rather than improving the scalability of our method. In the comparisons of our method with baseline methods, all methods use the same number of queries at each iteration and we plot the loss against the query (or iteration) number, so the parallelization does nothing to the comparison.

---

### Official Review · AnonReviewer2 · 2019-10-24
**Official Blind Review #2**

**Rating:** 8

**Review:**

This paper proposed a novel learning to learn framework based on zeroth-order optimization. Specifically, the framework consists of three parts: (1) UpdateRNN for learning the parameter update rules (2) Guided gradient estimation and search (3) QueryRNN that dynamically adapts the Gaussian sampling rules for covariance estimation in (2).

Experimental results on generating adversarial examples from black-box machine learning models as well as a binary classification problem demonstrate improved performance over several existing baselines, such as better query efficiency or faster empirical convergence in the loss function. An ablation study is also conducted to study the effect of each component in the proposed framework.

Overall, this paper is pleasant to read and well-motivated. The applications are of practical importance. Given that the empirical results suggest faster convergence than the compared methods, it will be great if the authors can also discuss how to prove the improved convergence in theory.

*** Post-rebuttal comments
I thank the authors for the clarification.
***

**Experience Assessment:**

I have published in this field for several years.

**Review Assessment: Checking Correctness Of Derivations And Theory:**

I carefully checked the derivations and theory.

**Review Assessment: Checking Correctness Of Experiments:**

I carefully checked the experiments.

**Review Assessment: Thoroughness In Paper Reading:**

I read the paper thoroughly.

---

> ### Author Response · Authors · 2019-11-15
> **Response to Reviewer #2**
>
> Thank you for the suggestion! In fact, this is still an open problem for the learning to learn community. Even in the first-order case, none of the existing work is able to provide an improved convergence rate (or even the same convergence rate) for learned optimizers, compared to the hand-designed ones. This is an interesting future direction that we are currently pursuing. In our future work, we aim to prove the improved convergence in theory and we have added more discussion in the conclusion part.

---

### Official Review · AnonReviewer1 · 2019-10-24
**Official Blind Review #1**

**Rating:** 6

**Review:**

This paper proposed a learning to learn (L2L) framework for zeroth-order (ZO) optimization, and demonstrated its effectiveness on black-box adversarial example generation.  The approach is novel since L2L provides a new perspective to zeroth-order optimization.

However, I have some concerns about the current version.

1) The knowledge that one should use to train UpdateRNN and  QueryRNN is not clear. A clear presentation is required

2) Please clarify the optimization variables in (4). In general, the problem is not clearly defined.

3) Eq. 5 is a query-expensive gradient estimate. Will it make training extremely expensive?

4) The computation and query complexity are unclear during training and testing.

5) Pros and cons of L2L? It seems that training a L2L network is not easy. Does its advantage exist only when inference? A better discussion should be made.

########## post-feedback #######
Thanks for the response. In the pros of L2L, the authors mentioned "The learned optimizer is trained on a small subset of optimization problems and apply in a wide range of problems in similar classes." In the setting of attack generation, does it mean that there exists an attack transferbility from a small group of training images to a large group of testing images? Is the transferbility a requirement for applying L2L in design of attacks. Please try to make these points clearer in the revised version. I keep my decision 'weak accept'.

**Experience Assessment:**

I have published in this field for several years.

**Review Assessment: Checking Correctness Of Derivations And Theory:**

N/A

**Review Assessment: Checking Correctness Of Experiments:**

I assessed the sensibility of the experiments.

**Review Assessment: Thoroughness In Paper Reading:**

I read the paper at least twice and used my best judgement in assessing the paper.

---

> ### Author Response · Authors · 2019-11-15
> **Response to Reviewer #1 (2/2)**
>
> Thank you for your comments and suggestions! Here are our responses to your questions.
>
> Q5: Pros and cons of L2L? It seems that training a L2L network is not easy. Does its advantage exist only when inference? A better discussion should be made.
>
> Applying the L2L framework can lead to several promising advantages:
>  1. Improving hand-designed algorithms with learned optimization rules: Currently, when solving specific classes of ML problems, we have to select from a big family of optimization algorithms and laboriously tune the hyperparameters. However, we can only obtain the tuned algorithms to the best of our knowledge and it’s difficult and laborious for us to design a better one. The L2L framework enables us to use the learned optimizer to automatically learn the optimization rules tailored to specific classes of optimization problems by exploring the particular loss landscapes. Our experimental results have shown that the learned optimizer can outperform highly tuned hand-designed algorithms.
> 2. Generalization across specific classes of problems: The learned optimizer is trained on a small subset of optimization problems and apply in a wide range of problems in similar classes. To illustrate, in the black-box adversarial attack task, we only used 100 images (corresponding to 100 different optimization problems in this class) to train the optimizer and the learned optimizer could be applied to solve optimization problems on all other test images (serving as an online “attacker”). The improvements the learned optimizer brings about in the class of problems can outweigh the overhead of training in a small subset.
> 3. Generalization to different classes of problems: In our experiments, we proposed a simple finetune scheme for applying the learned optimizer to different classes of optimization problems (see Section 4.3) and the experimental results showed that the learned optimizer could learn a rather general optimization rules. It means that probably there’s no need to train the optimizer from scratch when applied in new classes of problems. It also leads to a possible future direction of the L2L framework: pretraining the optimizer on difficult optimization problems and applying in general classes of problems with or without finetuning.
>
> The main disadvantage of L2L is the difficulty to theoretically prove its convergence rate and extra training overhead (if the optimizer has to be trained from scratch). In our future work, we aim to prove the improved convergence in theory and further improve the training methodology. We have added more discussion in the conclusion part.

---

> ### Author Response · Authors · 2019-11-15
> **Response to Reviewer #1 (1/2)**
>
> Thank you for your comments and suggestions! Here are our responses to your questions.
>
> Q1: The knowledge that one should use to train UpdateRNN and QueryRNN is not clear. A clear presentation is required
>
> The performance of the RNN optimizer is measured by that of the optimizee whose parameter updates are proposed by the RNN optimizer. Thus, we can use the optimizee loss function as part of the objective function (as described in equation (4)) to directly train the parameters of both the UpdateRNN and the QueryRNN, which can be done by truncated BPTT. Note that we use zeroth-order optimization method for training the optimizer if the gradients of the optimizee are not available in the training stage either (as described in Section 3.3), the only knowledge we need to train the optimizer is function values rather than explicit gradients of the optimizee. So our method can be applied as long as the function values of the optimizee are available.
>
>
> Q2: Please clarify the optimization variables in (4). In general, the problem is not clearly defined.
>
> Sorry for the confusion. To define the objective function (4) more clearly, we rewrite the optimizee’s parameters \theta_t as \theta_t(\phi) since \theta_t is updated by the RNN optimizer as in equation (2) and thus determined by the RNN optimizer’s parameters \phi. In a slight abuse of notation, we can also rewrite the predicted covariance matrix \Sigma_t as \Sigma_t(\phi) since it is proposed by the QueryRNN as in equation (3). The exact optimization variables in equation (4) are the RNN optimizer’s parameters \phi which include the parameters of both the UpdateRNN and the QueryRNN. We have revised Section 3.2 in the paper to make it more clear.
>
>
> Q3: Eq. 5 is a query-expensive gradient estimate. Will it make training extremely expensive?
>
> For problems of high dimensions, the coordinatewise ZO gradient estimator in equation (5) does require function queries linearly scale with the problem dimension. But this estimator can be computed in parallel, the computational overhead would be released a lot. We have experimented with the MNIST attack task (the problem dimension is 784) to estimate the computation overhead. We compare two methods:
> 1) use equation (5) to approximate the optimizee gradient.
> 2) assume the gradient of the optimizee model is available at training time and use traditional backpropagation (note that this assumption is made in Chen et al. (2017b) but is usually not the case, so we only use this method as the baseline for comparison).
> We find that the training time of 1) is about twice that of 2), which is acceptable. Moreover, potentially there could be several approaches to further reduce training time, such as sampling d'<d dimensions to estimate the gradient in equation (5) at each iteration.
>
>
> Q4: The computation and query complexity are unclear during training and testing.
>
> In the training stage, at each step of the forward pass, the optimizer uses equation (1) to obtain ZO gradient estimator whose query complexity is O(q) where q is the query number. In the backward pass, to backpropagate through the optimizee model, we apply coordinatewise ZO gradient estimator in equation (5) to approximate its gradient and its query complexity is O(d) where d is the problem dimension. Thus, for each step of the training stage, the total query complexity is O(q+d). If the gradient of the optimizee is available during training (which is the assumption made in Chen et al. (2017b)), we can use direct backpropagation instead of gradient estimation to train the zeroth-order optimizer, so the query complexity reduces to O(q). Note that both equation (1) and (5) can be computed in parallel to reduce computation time.
>
> In the testing stage (when using the optimizer to solve a given optimization problem), we only need to compute equation (1) for the forward pass. So for each step, the total query complexity is O(q). This is the same with other hand-designed ZO optimizers.
>
> For the computation complexity, most of the ZO optimizers (including the proposed one) have time complexity proportional to the number of queries, so the same observations will hold also for computation complexity.

---

### Decision · Program_Chairs · 2019-12-19

**Decision:**

Accept (Poster)

**Comment:**

This paper proposes to extend learning to learn framework based on zeroth-order optimization. Generally, the paper is well presented and easy to follow. The core idea is to incorporate another RNN to adaptively to learn the Gaussian sampling rule.  Although the method does not seem to have a strong theorical support, its effectiveness is evaluated in the well-organized experiments including realistic tasks like black-box adversarial attack.
All reviewers including two experts in this field admit the novelty of the methods and are positive to the acceptance. I’d like to support their opinions and recommend accepting the paper.
As R#1 still finds some details unclear, please try to clarify these points in the final version of the paper.